# Intrusion Detection in Vehicle Controller Area Network (CAN) Bus Using Machine Learning: A Comparative Performance Study

**DOI:** 10.3390/s23073610

**Published:** 2023-03-30

**Authors:** Bifta Sama Bari, Kumar Yelamarthi, Sheikh Ghafoor

**Affiliations:** 1Department of Electrical and Computer Engineering, Tennessee Tech University, Cookeville, TN 38501, USA; 2Department of Computer Science, Tennessee Tech University, Cookeville, TN 38501, USA

**Keywords:** vehicle security, cyber-physical system, CAN, intrusion detection, machine learning

## Abstract

Electronic Control Units (ECUs) have been increasingly used in modern vehicles to control the operations of the vehicle, improve driving comfort, and safety. For the operation of the vehicle, these ECUs communicate using a Controller Area Network (CAN) protocol that has many security vulnerabilities. According to the report of Upstream 2022, more than 900 automotive cybersecurity incidents were reported in 2021 only. In addition to developing a more secure CAN protocol, intrusion detection can provide a path to mitigate cyberattacks on the vehicle. This paper proposes a machine learning-based intrusion detection system (IDS) using a Support Vector Machine (SVM), Decision Tree (DT), and K-Nearest Neighbor (KNN) and investigates the effectiveness of the IDS using multiple real-world datasets. The novelty of our developed IDS is that it has been trained and tested on multiple vehicular datasets (Kia Soul and a Chevrolet Spark) to detect and classify intrusion. Our IDS has achieved accuracy up to 99.9% with a high true positive and a low false negative rate. Finally, the comparison of our performance evaluation outcomes demonstrates that the proposed IDS outperforms the existing works in terms of its liability and efficiency to detect cyber-attacks with a minimal error rate.

## 1. Introduction

Modern vehicles have been at the frontline of the advancement of automotive technology in recent years [1]. As a result, today’s automobiles are becoming more perceptive and delivering a more comprehensive range of practical, cutting-edge applications that cover many functionalities. Hundreds of Electronic Control Units (ECUs) are used to control these features, and they are all linked together by employing a CAN bus. ECU controls and monitors a vehicle’s subsystem to improve energy efficiency and reduce vibration as well as noise [2].

The increasing usage of ECU in automotive systems has led to significant improvement of functionalities. Although these advancements have made our lives easier, they have also made vehicles easier targets for cyber-attacks [3]. CAN lacks security including encryption and authentication to protect communication from cyber threats [4]. It has been demonstrated by researchers that in-vehicle networks have serious security vulnerabilities [5]. Injecting fake messages into CAN bus and manipulating and reading an ECU through vulnerable interfaces are two examples in which an attacker can take physical access to a vehicle. The Compact Disc (CD) players, Universal Serial Bus (USB), and On-Board Diagnostics (OBD)-II are the examples of vulnerable interfaces. Moreover, with the advancement of wireless technologies, including Bluetooth, Radio, Wi-Fi, Cellular, Long-Term Evolution (LTE), and 5G, vehicles are becoming extremely advanced with the help of these technologies, and they can now interact with their surroundings [6]. For example, vehicle key fobs have been used to successfully hack a live system. Additionally, ECUs can receive any ECU-to-ECU broadcasting signals on the same bus, and they have no way of knowing who sent them. It has been demonstrated how malicious attacks, such as packet injection and data manipulation, can cause fake packets to confuse essential parts guaranteeing drivers’ safety [7]. There are some other ways of vehicular attacks, including Radio, Tire Pressure Monitoring Systems (TPMS), GPS, Electronic Windows, and hacking of steering and brakes [8,9]. Thus, vehicular attacks are harmful to both vehicle and the driver since they create a lifetime risk for the driver [10]. Therefore, it is essential to detect intrusion on the vehicle, which could save the vehicle from being damaged as well as preserve human lives [11].

A vehicle can be affected by numerous sorts of attacks. For example, Denial-Of-Service (DoS) attacks, flooding attacks, fuzzy attacks, spoofing attacks, malfunction attacks, close proximity vulnerabilities, sybil attacks, replay attacks, routing attacks, remote sensor attacks, and impersonation attacks [12]. Several studies have investigated inter- and intra-vehicular communications safety issues [13]. For example, intrusion detection sensors are gaining more and more attention because of how effectively and easily they can detect intrusions [14,15]. An approach with an Advanced Encryption Standard (AES) cipher was developed by Noura et al. [16] that provides data confidentiality with minimum resources. It reduces computations, power, and memory. Castiglione et al. [17] proposed an approach utilizing lightweight block ciphers to secure in-vehicle communication with limited hardware and software resources. Mundhenk et al. [18] proposed a system called a Lightweight Authentication for Secure Automotive Networks (LASAN) that secures vehicle communication with low computational resources, for instance, power and network bandwidth. These IDS may be effective only for specific threat models which have already been considered in the design stages [19].

The majority of existing work on CAN protocol security has focused on physical factors, like limiting access controls and encrypting CAN communication [20]. However, there is still a need to develop a more efficient IDS. Indeed, the efficiency of CAN bus communications will be diminished due to physical access limitations. Cryptography is not always effective with such a lightweight system. To cope with the problem of conventional communication networks, ML-based IDS techniques are employed. The goal is to record the fundamental statistical properties of information and employ them in the detection of any kind of attack. Intrusion detection methods, using an SVM, DT, Multilayer Perceptron (MLP), and Random Forest (RF) are developed for classifying attack types [21]. Since the computational power of the conventional ECU is limited to the handle of such a complex procedure, ML algorithms are utilized for a vehicular network.

Promising and effective outcomes in solving complicated challenges, for instance, automatic system diagnostics and identification [22], fault detection in wireless system [23,24], cyber threat detection [25], and specific security problems in other fields have been generated using ML tools throughout the past decade [26,27]. To detect intrusions, ML approaches can be highly effective. However, a broadly recognized framework or model for identifying cyber-attacks that can be established and classified consistently is still lacking [7]. They can be improved by utilizing other ML models and a large amount of CAN dataset. This encourages us to investigate the possibilities of using efficient ML techniques, such as SVM, DT, and KNN, with a large amount of CAN datasets to overcome the current security concerns with in-vehicle CAN buses. In addition, most of the existing models for intrusion detection are employed only for a specific vehicle dataset. There is no explanation of the efficacy of the system when applied to another vehicle CAN data to the model. Thus, two different datasets are used in our proposed work to develop the system’s reliability.

The objective of this research is to develop an IDS based on ML techniques that can be applied to vehicles to secure the in-vehicular network, for example, the CAN network. This paper investigates the efficiency of intrusion detection using multiple datasets that are collected from real-time vehicles. The proposed approach trains on high-dimensional CAN packet data to identify the statistical differences between normal and attack packets after the dimension reduction. It extracts the corresponding features to classify the attack. Firstly, three ML supervised classifiers including SVM, DT, and KNN were used to classify attacks, including DoS, fuzzy, impersonation, and attack-free state using the Kia Soul car dataset (Dataset 1). Once satisfactory detection outcomes were achieved, another dataset (Dataset 2) from the Chevrolet Spark car was employed to evaluate the model performance and classify fuzzy, flooding, malfunction attacks. In both datasets, essential features are extracted for reducing the system complexity and computational time. Then, a comparative analysis was performed with the ML outcomes. The next step of this work is to implement the ideas on a real-time test bed and compare the performance. The key contributions of our work are as follows:A critical review of existing vehicular IDS to identify the research gap and develop an efficient IDS using ML.To the best of the authors’ knowledge, this work is the first studying multiple datasets collected from real vehicles (a Kia Soul car and a Chevrolet Spark car) to detect and classify intrusion in the vehicle.To develop an ML-based CAN bus IDS using three classifiers: SVM, KNN, and DT.Attacks detected: DoS, fuzzy, flooding, impersonation, malfunction, and attack-free state.Essential feature extraction to reduce system complexity and computational time.To achieve a high true positive rate and a low false negative rate.

Overall, this paper represents an effort to understand the necessity of cyber-attack detection on vehicles and thus develop a safer and sturdier system for intrusion detection. Experimental outcomes exhibit that the proposed work yields superior classification accuracy with small computation complexity and computational time. The rest of the paper is organized as follows: Section 2 presents a brief overview of the vehicular function and related work with existing research gaps on vehicular security. Section 3 defines the proposed methodology and explains each contribution in detail. Experimental results are presented in Section 4. Performance analysis of the experimental outcomes with possible future recommendations to mitigate current issues are presented in Section 5, followed by a conclusion in Section 6.

## 2. Background and Critical Review of the Existing Research Gap

### 2.1. Background

ECU

In modern electric vehicles, ECU is a device that controls specific functionality in the vehicles, including braking, airbag and engine control, parking, and door lock/unlock system [7]. A modern vehicle contains up to 70 ECUs. These ECUs are employed to control sensors and actuators [28,29]. A low latency and compact design system were developed for direct Vehicle-to-Vehicle (V2V) and Vehicle-to-Infrastructure (V2I) communication in [30,31]. In a vehicular network, ECUs can communicate with other ECUs. Each ECU consists of a CAN controller, CAN transceiver, and a microcontroller [32]. Several researchers have sent out fake messages to different ECUs utilizing the in-vehicle networks, peruse ECU memory, and ECU security keys. Such attacks on ECU can cause severe repercussions on the vehicle system and greatly harm drivers’ safety.

CAN bus

CAN is a broadcast protocol that can handle a baud rate of up to 1 Mb/s on a single bus [11]. It is widely utilized to in-vehicle networks because it can reduce wiring costs, weight, complexity, and higher speed. In addition, for a reliable transmission and fast recovery, it ensures robustness with an efficient error detection mechanism [33]. A microcontroller is connected to the CAN controller that has two pins, the transmitter and receiver. The CAN transceiver drives and detects data communication to and from the bus. The differential voltages are outputs and consist of 2 states of voltages, dominant (or logical 0) and recessive (logical 1). The two pins on the CAN transceiver are connected directly to the bus, allowing the ECU to transmit and receive a message from the bus [32].

CAN protocol is comprised of different abstraction layers including physical and transfer layers [34]. It has two types of physical layer standards, low speed and high speed. The low-speed standard (CANL) requires a single-wired bus and devices that self-terminate by 120 resistors ohm on the CAN Bus [35]. A two-wired half duplex serial network technology is used to build a high-speed CAN bus (CANH). The transfer layer abstraction receives messages from the physical layer and transmits those messages using the CAN bus. This layer is responsible for message framing, timing synchronization, fault confinement, arbitration, error detection, acknowledgment, and signaling. These properties allow safe message communication between ECUs.

### 2.2. Different Intrusions on Vehicle

There are different types of intrusion that occur in the vehicle [36,37]. Common types of attacks are GPS spoofing attacks, remote sensor attacks, location, trailing attacks, close proximity vulnerabilities, software flashing attacks, integrated business services attacks, DoS attacks, routing attacks, replay attacks, impersonation attacks, falsified-information attacks, fuzzing attacks, flooding attacks, sybil attacks, sniffing attacks, and malfunction attacks [38,39,40,41]. Remote sensor attacks are caused on the vehicle’s camera, radar, and other sensors. GPS spoofing attacks are caused by sending fake messages. Bluetooth, TPMS, key fob, and keyless entry are the ways to cause close proximity vulnerabilities [42]. Software flashing methods are caused by gear shift, servo steering, ignition system, and electronic window lift [43] Remote telediagnosis, entertainment, and remote software update occur in integrated business services attacks [44]. In a DoS attack, authentic users cannot access network services and correct messages cannot reach their destinations. Replay attacks usually happen in some authorization and key agreement protocols. The adversary can send or broadcast false information and safety alerts in a falsified information attack. Message tampering, suppression, fabrication, and alteration may produce a fake message. An impersonation attack can be implemented by using another identity or a fake identity [12].

### 2.3. Related Study with Research Gaps

There is a strong need to secure vehicle communication since it directly impacts the security of vehicles and the lives of their occupants, including drivers and passengers. It is difficult to accomplish with the CAN protocol due to the limitations of CANs, for example, the network’s susceptibility to attacks including DoS, impersonation, and fuzzy attacks. As a result, it has led to innovation with a challenging opportunity to solve the problem of creating an IDS for such a network. Indeed, several studies have been conducted to address this issue. The most essential related studies investigating IDS for vehicular communication are discussed in the following with existing research gaps.

Moulahi et al. [21] proposed malicious intrusion detection in vehicles using the CAN bus protocol. RF, SVM, MLP, and DT classifiers are employed to distinguish between normal and malicious communications. DoS, impersonation, and fuzzy attack have been detected in their approach. The four ML techniques are able to successfully detect the attacks. However, in this system, only known attacks are detected and high computational resources are required. Thus, it could be upgraded by detecting unknown or new intrusions. In addition, a large number of datasets should be used in their model.

Han et al. [14] developed an attack identification technique based on the event-triggered characteristics of the CAN IDs depending on the vehicle model. The four attacks have been detected: flooding, fuzzy, malfunction, and replay. The performance of the proposed method has been evaluated by evaluating the accuracy, cost, and time depending on the defined time window for specific attack types. It is a real-time application. However, the tree-based ML model’s accuracy should be increased.

Lee et al. [11] analyzed the offset ratio and the time interval between the request and the response working on a remote frame and data frame to create an IDS. This response is used to analyze the behavior if it is an attack (intrusion detection) or normal behavior. Three types of attacks including DoS, fuzzy, and impersonation attacks are detected in CAN-based networks. However, a metric like the accuracy of attack detection is not given to determine whether or not the proposed approach achieved the best detection performance. By using the same dataset employed in [11], Tariq et al. [45] used Recurrent Neural Network (RNN) and heuristics to detect three types of attacks, including DoS, replay, and fuzzy attacks. The authors used both Neural Networks (NN) and network traffic signatures. The accuracy of intrusion detection is high, however, these authors did not consider the technique for unseen attacks.

Miller et al. [46] proposed a car hacking system using Wi-Fi, radio data, and telematics. It exposed the overall hacking technique, which helps vehicle security researchers. It can hack the car by deactivating its steering and brake function. It is an excellent way to hack the vehicle remotely. However, this approach must be validated by applying the experiment to the new and updated vehicles.

Groza and Murvay [15] developed an IDS based on bloom filtering. This filter has no false negatives and provides 100% recall rate. This filtering method is utilized based on frame identifiers and part of the data fields to test frame periodicity. It facilitates the detection of frame modification attacks or possible replays. This approach can also be used with other types of in-vehicle communication. The limitation of this approach is that it included an essential overload on ECU, which could affect their timely response.

A Deep Neural Network (DNN)-based system was developed for intrusion detection in CANs [7]. Deep learning (DL) techniques are used to distinguish between normal behavior and attacks. The comparison between DNN-based IDS and standard NNs shows that a DNN is better in terms of accuracy with a real-time response.

Wu et al. [13] developed a novel IDS based on the information entropy. In this method, optimizing the decision conditions and enhancing the sliding windows help to enhance intrusion detection accuracy while decreasing the false positive rate. Furthermore, the effectiveness of the proposed method was demonstrated in an experimental study providing real-time responses to intrusion with the accuracy of 100% for DoS and 92.3% for injection attack. It reduces automotive costs and computing performance. Although the studies found encouraging outcomes, the authors did not consider how the vehicle’s operational state could affect information entropy.

Mundhenk et al. [18] proposed an approach that secures vehicle communication with low computational resources, for instance, power and network bandwidth. It uses an Advanced Encryption Standard (AES)-based cipher that uses 128 bits. The outcome of this study is compared with two existing authentication frameworks, Transport Layer Security (TLS) and Timed Efficient Stream Loss-Tolerant Authentication (TESLA). This system performs highly efficient lowering latencies in typical network sizes of 60 to 100 ECUs by an average factor of 49 for TESLA and factor of 234 for TLS. However, it is based on a software-oriented cipher. The integration of hardware and software ciphers could be employed to implement more secure communication.

Castiglione et al. [17] proposed a technique utilizing lightweight block ciphers to secure in-vehicle communication with limited hardware and software resources. The outcomes of their study reveal that this concept ensures a high level of safety with a low delay while having a negligible effect on the vehicle’s performance. The three most well-known lightweight ciphers have been utilized, namely SIMON, SPECK, and PRESENT. It is a progressive work proposed by Mundhenk et al. [18] that is completely software-oriented. Compared to the study [18], it is more appropriate to utilize hardware-oriented ciphers than software-oriented ciphers. Thus, using hardware-oriented ciphers (SIMON) is more effective than software-oriented ones (SPECK). However, the delay accumulated during the encryption or decryption phase of the messages depends not only on the algorithm used but also on the system’s hardware. Hence, this approach still has an issue due to hardware with really low performance.

Table 1 represents a summary of the existing intrusion detection methods in a vehicle with their strengths and current research gap.

Several studies have been carried out on the issue of implementing an IDS in CAN-based networks; nonetheless, there is still a need for improvement. Most of the prior research either focuses on the behavior of exchanged frames or makes minimal use of the data contained within the frames. Moreover, conventional methods of classification are not always suitable for modern vehicles. In addition, a specific vehicle dataset is utilized to classify and test with only a specific developed classification model, which could raise a reliability issue. Moreover, there is a lack of real-time vehicle data to establish the model. To ensure the vehicles and their occupant’s safety, including passengers and drivers, this paper aims to develop an efficient IDS for a vehicular CAN bus by performing a thorough data analysis collected from two types of real vehicles. In addition, three supervised ML methods, including SVM, DT, and KNN are performed to detect and classify intrusion effectively containing fifteen essential features.

## 3. Methodology

In order to develop the proposed IDS, a variety of ML approaches have been implemented using Python. Figure 1 shows the overall system architecture of the proposed work.

The details of the proposed system architecture are discussed in the following:

### 3.1. Data Description

In this study, two types of real vehicle CAN bus intrusion datasets are utilized. Dataset 1 is collected from a Kia Soul [35] and Dataset 2 is collected from a Chevrolet Spark [48] car. The description of each dataset is described as follows.

#### 3.1.1. Dataset 1

Dataset 1 consists of three types of attacks including DoS, fuzzy and impersonation attack, and attack-free state. A dataset of 461,341 examples is employed that contains 65,657 DoS attacks, 59,198 fuzzy attacks, 99,547 impersonation attacks, and 236,939 attack- free datasets. These in-vehicle datasets are extracted from the Kia Soul car. Datasets were constructed by logging CAN traffic via the OBD-II port from a real vehicle while message injection attacks were performed [35]. The three types of attack scenarios including DoS attack, fuzzy attack, and impersonation attack, are presented in the Figure 2.

DoS attack

Injecting messages of 0 × 000 CAN ID in a short cycle. In a short bus cycle, an attacker can inject many high-priority messages. There are three injected 0 × 000 messages between request and response messages. Since all of the nodes are connected to the same bus, an increase in traffic could delay the delivery of other messages or even prevent the system from responding to the driver’s requests altogether.

Fuzzy Attack

Injecting messages of spoofed random CAN ID and DATA values. An attacker can inject malicious data into the body text of a randomly faked identification. This results in lots of functional messages being sent to all nodes, leading to unexpected vehicle behaviors. In the fuzzy attack, the attacker monitors vehicle-to-vehicle communications and carefully selects target identities in order to make false behaviors.

Impersonation Attack

Injecting messages of impersonating node, arbitration ID = 0 × 164. In order to stop messages from being sent, an attacker must have control of the target node and either establish or manipulate an impersonating node. If the victim node suddenly stops sending data, all of its messages from target node will be deleted from the bus. A data frame transmission is expected to occur instantly after an ECU receives a remote frame. An existing node is attacked or broken if the receiving node does not receive a response to a remote frame. An attacker can then plant a impersonating node to reply to a remote frame. This means that the impersonating node will periodically send out data frames and respond to a remote frame in the same way that the target node did.

Attack-Free State

Normal CAN messages.

#### 3.1.2. Dataset 2

Dataset 2 consists of three types of attacks including flooding, fuzzy and malfunction attack, and attack-free state. A dataset of 313,930 examples is employed that contains 84,999 flooding attacks, 40,999 fuzzy attacks, and 50,999 malfunction attacks. These in-vehicle datasets are extracted from the Chevrolet Spark car. Datasets were constructed by logging CAN traffic via the OBD-II port from a real vehicle while message injection attacks were performed. The three types of attack scenarios, including flooding attack, fuzzy attack, and malfunction attack are described here.

Flooding Attack

In a flooding attack, when a receiver ECU node receives CAN messages simultaneously from several sender ECU nodes, the values of the CAN IDs are verified to establish the order of acceptance. A CAN ID’s importance increases as its value decreases. The attack can disrupt normal driving by limiting connectivity between ECU nodes.

Fuzzy Attack

For the fuzzy attack, CAN messages are produced at random. Both the ID field and the data field went through this procedure. The randomly generated CAN ID ranged from 0 × 000 to 0 × 7FF and contained CAN IDs that were taken from the car as well as CAN IDs that were not.

Malfunction Attack

In the malfunction attack, one of the vehicle’s extractable CAN IDs is chosen at random. Malfunction attacks involve both the manipulation of the data field and the injection of CAN IDs chosen at random. When the values in the data field consisting of 8 bytes were manipulated using 00 or a random value, abnormal behaviors are found on the vehicle.

Attack-Free State

An attack-free state represents all the regular CAN bus traffic data. Embedded sensors and devices that aid vehicle operation receive their status updates from the CAN IDs over the CAN Bus.

### 3.2. Data Pre-Processing and Feature Extraction

All the datasets are attributed as Timestamp, CAN ID, DLC, DATA [0], DATA [1], DATA [2], DATA [3], DATA [4], DATA [5], DATA [6], DATA [7]. The brief explanations are as follows:Timestamp: It is the recorded time (s);CAN ID: It is an identifier used to identify CAN message in HEX (ex. 043f);DLC: It is a number of data bytes ranging from 0 to 8;DATA [0~7]: It represents the data value (byte).

The procedure of data labeling has been carried out by performing prepossessing in accordance with the description of the dataset that was provided in [11]. After that, fifteen essential features are extracted from each dataset which contains adequate information. A large number of data used in the ML classification may delay the overall performance due to the large execution time. It also creates system complexity. Here, feature extraction has been performed to reduce the system complexity and execution time. The fifteen features are Timestamp, Last remote frame Timestamp, Frame ID, Previous frame ID, ID of previous of previous of frame ID, ID of previous of previous of previous of frame ID, Size of data filled in the frame, First byte to eighth byte of data (described in Table 2).

### 3.3. ML-Based Classification

The process of feature extraction is necessary before moving onto the classification. In ML, there are typically three different models that can be used for prediction: the regression model, the classification model, and the clustering model. The classification-based model or the clustering-based model can be used for real-time or predictive intrusion detection [49]. The classification-based model is utilized in the event of a supervised problem, whereas the clustering-based model is taken into consideration in the event of a non-supervised problem [21]. The main difference between supervised and unsupervised learning is that supervised learning uses labeled input and output data, while an unsupervised learning algorithm does not. Three popular supervised ML techniques are used such as SVM, DT, and KNN in this study. The purpose of using supervised ML classification instead of unsupervised classification is that the known labeled dataset is used in the supervised classification problem [38,45]. The selected dataset has been fed into three types of ML classifiers and the performance is measured based on the validation metrics. Before feeding to the ML classifier, the total data for each dataset are split into two types: training dataset and testing dataset. At first, 75% of the overall data are considered as the training dataset. The remaining 25% are used as a test dataset to test the classification model. Three ML classifiers are described in brief.

#### 3.3.1. SVM

SVM is an important and easy to use supervised ML technique for regression and classification [50]. In both linear and nonlinear systems, it is most beneficial for classification, but it can also be helpful when used for regression. This method employs Kernel functions to transform a low-dimensional feature into a high-dimensional feature in a non-linear system. The Gaussian Kernel, Radial Basis Function (RBF), Linear Kernel, Sigmoid Kernel, and Polynomial Kernel are the most popular kernel functions. It is built with a determination of decision boundaries. It utilizes labeled samples and can be generalized by the calculation of a learning model. Basically, it aims to locate a hyperplane within a finite number of data dimensions that may be used to accurately classify the training dataset (Figure 3). The test samples are compared to the hyperplane to determine the correct sample category during the testing. The straight line is the optimal hyperplane which is used to represent the maximum margin hyperplane. The dotted lines are presented to develop the maximum margin or best margin in which the vectors are placed for each class which is nearest to the optimal hyperplane. SVM has been effectively implemented in a wide variety of uses, such as image categorization, fault detection, and handwritten identification [51].

#### 3.3.2. DT

DT is a method of supervised learning that can be used to address challenges of classification or regression [21]. The classifier is organized like a tree, with internal nodes standing for the features of a dataset, branches used for the rules to make judgments, and leaf nodes which exhibit the final outcome. Decision node and leaf node are the two types of nodes used in a DT (Figure 4). It is structured like a tree which starts from the root node and further branches expand. Leaf nodes are the results of previous decisions and do not contain any additional branches, whereas decision nodes feature many branches that are used to make the actual decisions. A decision or test is made based on the features of the dataset.

#### 3.3.3. KNN

To handle problems of classification and regression, KNN is utilized as a supervised ML method [32]. For classification, at first, it calculates the number of nearest neighbors. Then, calculate the distance of testing observations with all training data using Euclidean distance. After that, it selects possible shortest distance observations from the testing point and calculates the probability of all shortest observations. In order to properly categorize a new data point, it makes use of all of the training data previously obtained. It assigns testing observation with the highest priority. During the training phase, it collects data that are similar to each other, and then it uses that data during the testing phase. Initially, the training data and a test instance are measured for various distances. Then, the class of the test dataset is predicted through using majority voting among the K-nearest training sample (Figure 5).

### 3.4. Performance Evaluation Matrices

One of the essential requirements in developing a reliable classification model is measuring the ML model’s efficiency. Different metrics are performed to determine the performance or quality of the classification model. These metrics are referred to as performance evaluation metrics. In our study, six popular performance evaluation matrices for ML classification are used including Confusion matrix, Precision, Recall, Accuracy, F1 score, and Cohen’s Kappa score.

After classification, the performance of the methods is evaluated using the confusion matrix. It provides a visual representation of the outcomes of a classification model as true positive (*TP*), false positive (*FP*), true negative (*TN*), and false negative (*FN*) rates.

In a confusion matrix:*TP*: Correctly predicted positive observations by the model;*FP*: Negative observations that incorrectly predicted as positive by the model;*TN*: Correctly predicted negative observations by the model;*FN*: Positive observations that incorrectly predicted as negative by the model.

Precision is a measurement of how well the model can accurately identify the positive class. It is defined by Equation (1):(1)$$Precision=\frac{TP}{TP+FP}$$

Recall is a metric for evaluating a model’s performance in accurate predictions for all the positive data points in a given dataset. The false-positive results are not considered in recall. It is defined by Equation (2):(2)$$Recall=\frac{TP}{TP+FN}$$

The accuracy of a classification method is the most important measure of its efficacy. An accuracy score might range from 0 to 1, where 1 represents a perfect classification model. It is defined by Equation (3):(3)$$Accuracy=\frac{Correct\;Predictions}{Total\;Predictions}=\frac{TP+TN}{TP+TN+FP+FN}$$

An F1 score can range from 0 to 1. When the F1 score is 1, both precision and recall represent a perfect model performance. If precision or recall is 0, then the F1 score is 0. It is defined by Equation (4):(4)$$F1\;Score=\frac{2\times Precision\times Recall}{Precision+Recall}$$

The Cohen’s Kappa score is a statistical measure of data classification using ML. Kappa can range from 0 to 1. A value of 0 means that there is no accuracy of the classification model, and a value of 1 means that there is perfect accuracy of the classification. In most cases, anything over 0.7 is considered to be a very good score. It is defined by Equation (5):(5)$$Cohen\;Kappa\;Score=\frac{P_{O}-P_{e}}{1-P_{e}}$$
where, *P_o_* is the observed accuracy, and *P_e_* is the expected accuracy.

The performance metrics are essential for evaluating how effectively a model is performed for the corresponding dataset. After training with intrusion data, several attacks are detected and classified successfully with superior performance than existing related works. The experimental results for each of the ML classifiers are presented in the next section.

## 4. Experimental Results

For Dataset 1, a DoS, fuzzy and impersonation attack, and no attack were detected. Similarly, flooding, fuzzy and malfunction attack, and no attack were detected in Dataset 2. In this experiment, there is a common attack detected called fuzzy attack. Here, five different types of attack, namely DoS, fuzzy, impersonation, flooding, malfunction, and attack-free vehicle were identified and classified. The outcomes of the experiment have been explained in this section for individual detection classes in both Dataset 1 and Dataset 2. The fifteen features used in this work which are extracted from each attack have been listed in the Table 2.

After important feature extraction, data classes are labeled according to each attack class and then fed into the ML classifiers including SVM, DT, and KNN for both individual dataset, Dataset 1, and Dataset 2. Then, the results are obtained for each ML technique by considering different performance evaluation matrices, such as Confusion matrix, Precision, Recall, Accuracy, F1 score, and Cohen’s Kappa score. The experimental outcomes are represented in the following sub-sections.

### 4.1. Experimental Results of Dataset 1 (KIA Soul Car)

DoS attacks, fuzzy attacks, impersonation attacks, and attack-free classes are identified and classified in Dataset 1 using SVM, DT, and KNN. The detection performances are shown below:

#### 4.1.1. SVM

The confusion matrix for SVM testing is presented in Figure 6. The confusion matrix of SVM clearly exhibits the correct and incorrect attack class detection, which helps to demonstrate the true positive and false negative rate of intrusion detection. Here, the Precision, Recall, Accuracy, F1 score, and Cohen’s Kappa score of overall intrusion detection using SVM are 0.975, 1.0, 0.975, 1.0, and 0.961, respectively. The following Table 3 shows the results for each individual attack using SVM in terms of performance evaluation matrices, and training and testing time.

#### 4.1.2. DT

The confusion matrix for DT testing is presented in Figure 7. Here, the Precision, Recall, Accuracy, F1 score, and Cohen’s Kappa score of overall intrusion detection using DT are 0.994, 1.0, 0.994, 1.0, and 0.990, respectively. Table 4 shows the results for each individual attack using performance evaluation matrices, and training and testing time.

#### 4.1.3. KNN

The confusion matrix for KNN testing is presented in Figure 8. Here, the Precision, Recall, Accuracy, F1 score, and Cohen’s Kappa score of overall intrusion detection using KNN are 0.9643, 1.0, 0.964, 1, and 0.945, respectively. The following Table 5 shows the results for each individual attack using KNN in terms of performance evaluation matrices, and training and testing time.

### 4.2. Experimental Results of Dataset 2 (Chevrolet Spark Car)

Dataset 2 includes detection and classification techniques for flooding, fuzzy, malfunction attack, and attack-free class. The classification abilities of SVM, DT, and KNN for Dataset 2 are described as follows:

#### 4.2.1. SVM

The confusion matrix for SVM testing is presented in Figure 9. The true positive and false negative rates of intrusion detection can be seen in the confusion matrix generated by the SVM. Here, the Precision, Recall, Accuracy, F1 score, and Cohen’s Kappa score of overall intrusion detection using SVM are 0.939, 1.0, 0.939, 1.0, and 0.912, respectively. Individual attack classification outcomes using SVM are evaluated by performance matrices. The classification performance, and training and testing time for SVM are shown in Table 6.

#### 4.2.2. DT

The confusion matrix for DT testing is presented in Figure 10. Here, the Precision, Recall, Accuracy, F1 score, and Cohen’s Kappa score of overall intrusion detection using DT are 0.999, 1.0, 0.999, 1.0, and 0.999, respectively. The following Table 7 shows the performance for each individual attack, and training and testing time using DT.

#### 4.2.3. KNN

The confusion matrix for KNN testing is presented in Figure 11. Here, the Precision, Recall, Accuracy, F1 score, and Cohen’s Kappa score of overall intrusion detection using KNN are 0.977, 1.0, 0.977, 1, and 0.968, respectively. The outcomes of each individual attack utilizing KNN are summarized in Table 8 with corresponding training and testing time.

The ML-based intrusion detection and classification performance have been analyzed and compared in the following section.

## 5. Performance Analysis and Future Recommendations

A CAN bus is the most popular vehicle communication technology and one of the most key components that must be protected from malicious threats. However, CAN buses lack adequate security because no safety precautions were taken during implementation, and CAN itself provides no defenses against malicious adversaries. Within this scope, an IDS offers additional protection that strengthens the vehicle’s security architectures.

In this study, an IDS is developed in which two totally different real vehicles of CAN bus datasets are used to detect and classify the vehicle intrusion using ML algorithms. In order to detect intrusions, three ML algorithms (SVM, DT, and KNN) are employed and fifteen important features are extracted. The IDS performance is evaluated over five sorts of vehicular attacks including DoS, fuzzy, flooding, impersonation and malfunction attacks, and attack-free states on real vehicular CAN bus datasets.

### 5.1. Performance Analysis of Overall Proposed IDS

While Dataset 1 is used to classify DoS, fuzzy, impersonation, and attack-free states, Dataset 2 is applied to classify flooding, fuzzy, malfunction, and attack-free states. Table 9 shows the performance comparison of three ML approaches used in our study.

The experimental findings for the employed ML classifiers (SVM, DT, and KNN) to detect intrusion are summarized in Table 9. In Dataset 1, DT achieved the highest levels of accuracy (99.4%) and Cohen’s Kappa score (0.990). When applied to Dataset 2, the corresponding values for accuracy and Cohen’s Kappa are 99.9% and 0.999, respectively. Figure 12 and Figure 13 depict the findings of comparing ML classifiers’ performance on Dataset 1 and Dataset 2 in terms of Precision, Recall, F1, and Cohen’s Kappa score, respectively.

Therefore, DT demonstrated the highest accuracy for intrusion detection in both Dataset 1 and Dataset 2. When compared to SVM and KNN on Dataset 1, DT’s accuracy is 1.9% and 3.1% better, respectively. According to Dataset 2, DT outperforms SVM and KNN in terms of accuracy by 6.3% and 2.25%, respectively. Figure 14 and Figure 15 show the accuracy comparison between the SVM, DT, and KNN for Dataset 1 and Dataset 2, in regard to overall intrusion detection performance.

Another contribution of the proposed method is an increase in the true positive rate and a decrease in the false negative rate of the intrusion detection system. Both the true positive and false negative rates of the proposed IDS are shown in Table 10.

When comparing Dataset 1 and Dataset 2, DT classifier has a high true positive rate which are 0.994 and 0.999, respectively. Furthermore, DT has a lower false negative rate than SVM and KNN for both datasets. To classify Dataset 2, DT achieves the best true positive rate whereas false negative rate is the lowest. Such a high degree of performance demonstrates that the proposed system could be beneficial for detecting vehicle intrusion.

### 5.2. Comparison of Different Attack Detection Performances

Precision values of 0.99, 1.0, 0.96, and 0.97 are obtained when feeding Dataset 1 into an SVM classifier for DoS, fuzzy, impersonation, and attack-free state classification, respectively. To a similar extent, DT classifier’s DoS, fuzzy, impersonation, and attack-free states all have precision values of 1, 0.99, 0.99, and 1.0, respectively. KNN, on the other hand, achieved a DoS detection precision of 0.99, a fuzzy precision of 0.99, an impersonation precision of 0.93, and an attack-free state precision of 0.97. Three ML classifiers are compared in terms of their Precision, Recall, and F1 score which are presented in Figure 16, Figure 17 and Figure 18 for DoS, fuzzy, and impersonation attacks, respectively.

Similarly, Figure 19, Figure 20 and Figure 21 evaluate the effectiveness of SVM, DT, and KNN for Dataset 2 with flooding, fuzzy, and malfunction attacks.

The performance evaluation matrices and execution times of the proposed study and relevant studies for vehicle intrusion detection are compared in Table 11.

The classification accuracy of the system is the most significant aspect of intrusion detection. As the error rate decreases, the effectiveness of the detection system increases. The higher detection accuracy ensures the applicability of the detection system with a lower error rate. When compared to the performance achieved by Moulahi et al. [21], our proposed work obtains a greater level of accuracy for SVM and DT with 0.23% and 1.23% for Dataset 1, respectively. In the case of Dataset 2, DT achieves a higher level of accuracy by 1.74% compared to [21]. While compared to [31], the accuracy that has been achieved using KNN for Dataset 1 is 1.74% greater than that of accuracy achieved in [32]. In addition, the F1 score of SVM and KNN of our proposed study is higher than [32] for both datasets. However, for the accuracy of SVM with Dataset 2 in our proposed study, it is a little bit lower than [21,32]. It could have happened because of the variation in the dataset, given that the amount of data in Dataset 2 that was utilized in our proposed study is almost 15% more than the amount of data that was used in [21,32]. As the amount of data varies more widely, the overall performance of SVM may be affected. Python programming was used for both the design and implementation of the proposed IDS. The programming was run on a Windows 11 computer with 3.6 GHz and 16 GB of RAM and an i7 processor. In addition, the Python program was executed in Google Colab as well as in a Jupyter notebook so that its performance could be observed. Hence, the intrusion detection performance is similar and there is only a negligible difference in execution time. On the other hand, the testing time required by KNN is significantly longer than that required by SVM and DT for both of the datasets employed in our proposed study. It happened because the KNN algorithm does more computation on test time rather than training time. In KNN, at the training stage, the algorithm stores a training set. During the testing stage, the algorithm searches for K-neighbors by using the data that were previously saved and comparing it to the sample that was used to classify the data. In addition, the proposed study makes use of more data than other relevant studies while still achieving higher accuracy, which maintains the efficacy of the intrusion detection. Nevertheless, the proposed work has a low error rate due to its high true positive and low false negative value. As a result, our proposed study for ML-based vehicle intrusion detection performs better than similar studies based on performance analysis and comparison of detection outcomes.

### 5.3. Uncertainties and Limitations with Future Recommendations

To the authors’ knowledge, this is the very first work to successfully develop an ML-based vehicle IDS employing multiple CAN datasets with a minimal error rate and high accuracy. In order to address the robustness of the proposed IDS developed by Dataset 1, Dataset 2 containing other attacks is applied to the developed ML models and achieved the desired outcomes. Still, it is essential to expand this research to enhance its applicability considering drawbacks that have also been identified. Some limitations of this study and existing IDS issues with the prospective solution to address these challenges are as follows:Misclassification issues often arise because of the similarities in attack behavior. More datasets containing similar attack characteristics used during training the network ought to be essential to overcome this issue. It is also recommended to apply deep learning algorithms that can classify data with slight differences in characteristics.In the widely used vehicle CAN dataset, including the datasets [34,47], there is a far difference between attack-free state dataset and attack dataset. Thus, a dataset in which all classes’ datasets are the same in their amount could be developed and applied to the ML model to boost up the overall classification efficiency.When a large amount of CAN data is applied in an ML-based IDS system, it could lengthen the training that leads to delay the classification process. In this case, a deep learning technique could be employed to deal with this issue since it can process a huge amount of datasets with the shortest execution time.Supervised ML classification techniques are used in our proposed IDS system and the systems proposed in [21,31] where only known attacks are detected. Therefore, an unsupervised classification method could be applied to investigate the detection performance using some new or unknown intrusions since unsupervised learning is a useful technique for data classification when a dataset lacks a label.

In our proposed work, we have not addressed test bed implementation to collect an intrusion dataset. In our future work, a simulated environment or a real-time test bed would be developed. It will not impact on the intrusion prediction performance. The predictive performance of the model is independent on simulation or implementation on a real-time test bed. Our study could also be extended by developing an advanced intrusion detection technique that can detect totally new and unknown intrusions. Moreover, the development of a deep learning technique could be an effective candidate to establish a more effective approach for intrusion detection in modern vehicles.

## 6. Conclusions

Due to the increased vulnerability, complexity, and diversity of modern vehicles, intrusion detection for vehicles have emerged as a crucial aspect in the security of automobile technology. This work examines whether or not a vehicle is under attack. The five most dangerous attacks on vehicles including DoS, fuzzy, impersonation, flooding, and malfunction attacks, and attack-free vehicles have been classified based on ML techniques. Three supervised learning including SVM, DT, and KNN have been utilized to classify the attacks for both Dataset 1 and Dataset 2 and compare their performances. Among the three ML techniques, DT outperforms SVM and KNN in terms of performance evaluation matrices and computational time. One of the most notable contributions of our work was to apply two different datasets to classify and test the intrusion detection model and achieve a small false negative and large true positive rate. The proposed work will eventually be applied to various in-vehicle communication safety purposes. Therefore, by considering the multiple CAN datasets, low error rate and high detection accuracy, our proposed method is more effective than other similar techniques for vehicle intrusion detection.

## Figures and Tables

**Figure 1 sensors-23-03610-f001:**
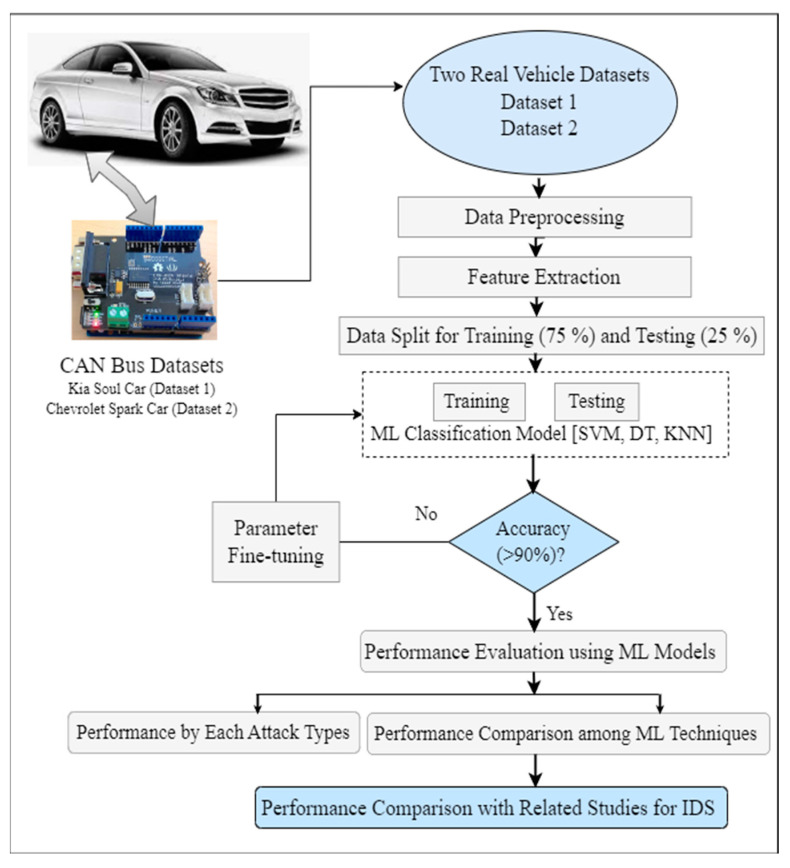
Architecture of the proposed ML-based IDS.

**Figure 2 sensors-23-03610-f002:**
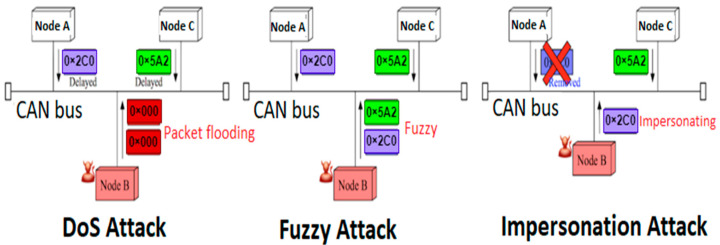
Three types of attack scenarios on a CAN bus.

**Figure 3 sensors-23-03610-f003:**
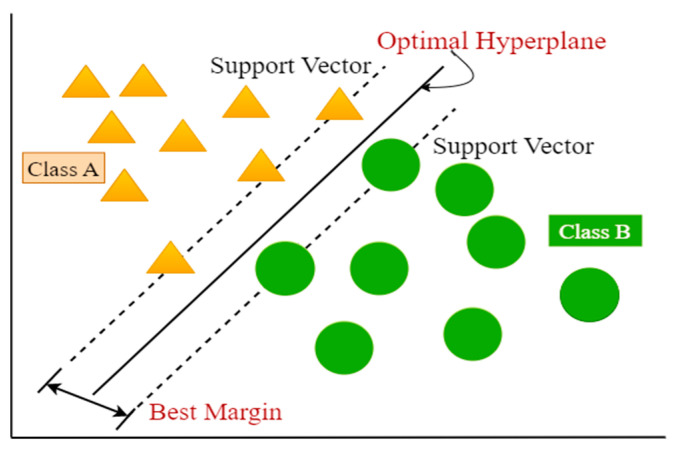
Basic structure of SVM.

**Figure 4 sensors-23-03610-f004:**
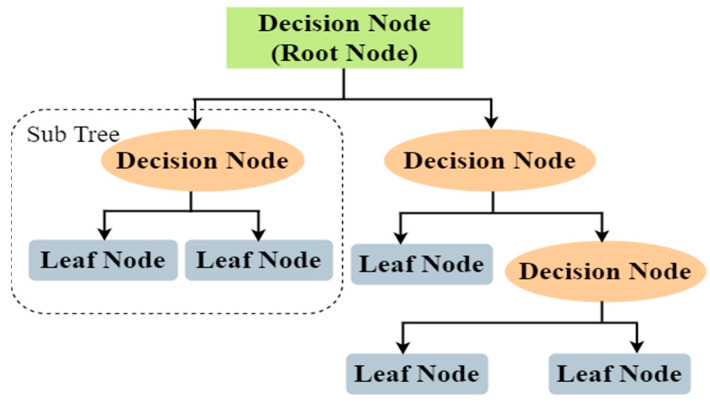
Basic structure of DT.

**Figure 5 sensors-23-03610-f005:**
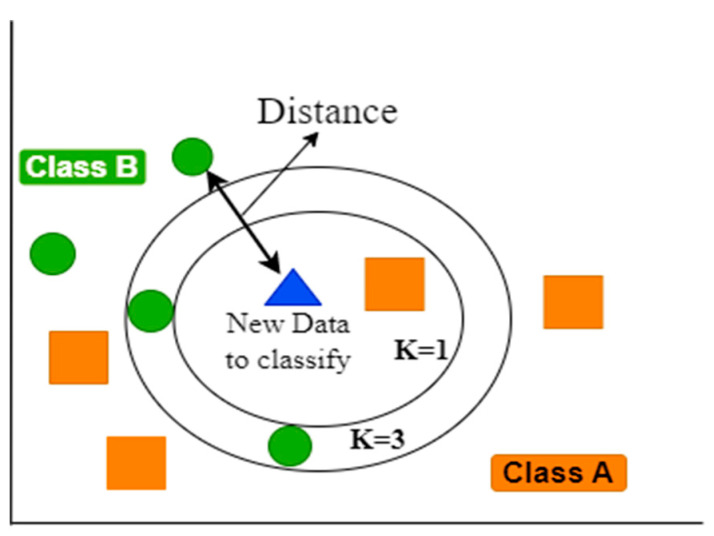
Basic structure of KNN.

**Figure 6 sensors-23-03610-f006:**
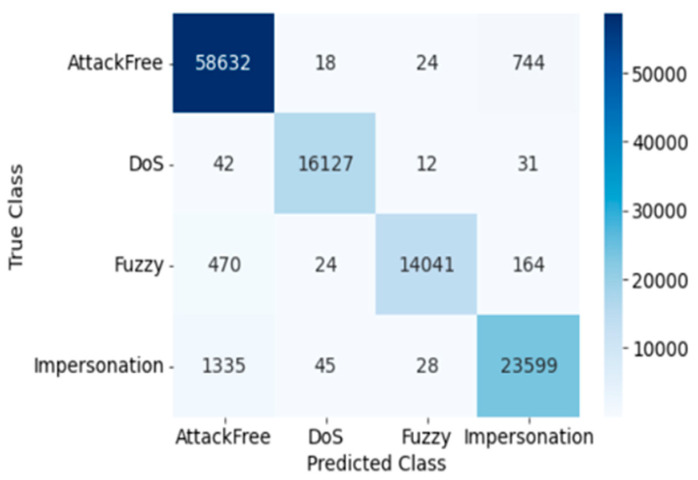
Confusion Matrix of SVM (Dataset 1).

**Figure 7 sensors-23-03610-f007:**
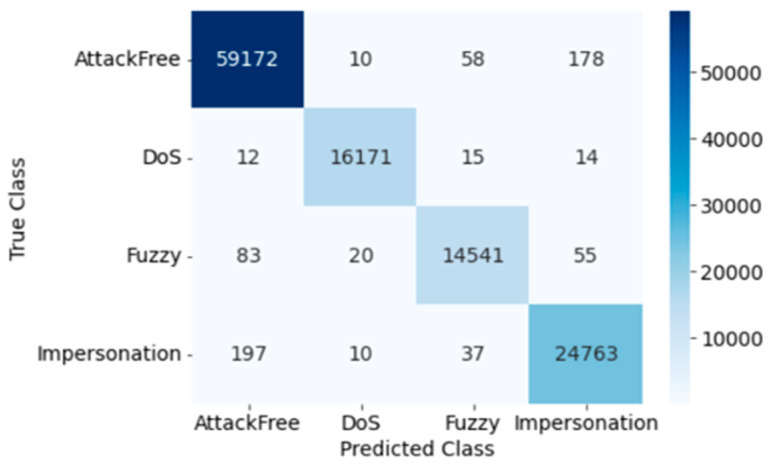
Confusion Matrix of DT (Dataset 1).

**Figure 8 sensors-23-03610-f008:**
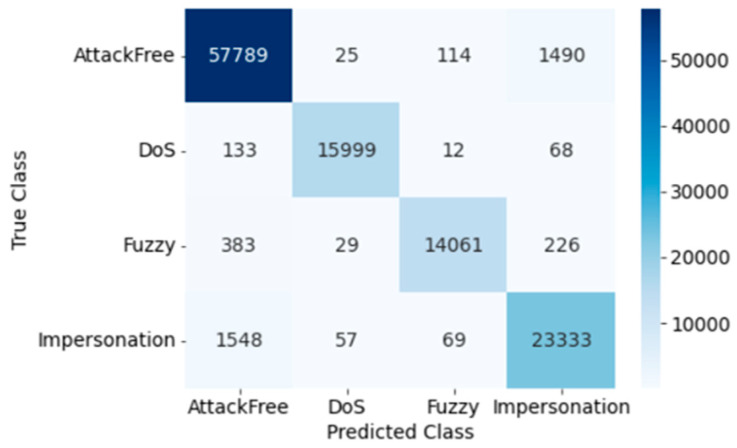
Confusion Matrix of KNN (Dataset 1).

**Figure 9 sensors-23-03610-f009:**
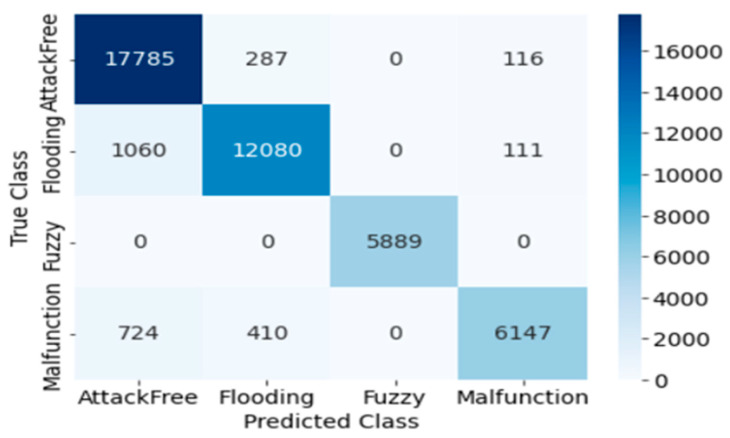
Confusion Matrix of SVM (Dataset 2).

**Figure 10 sensors-23-03610-f010:**
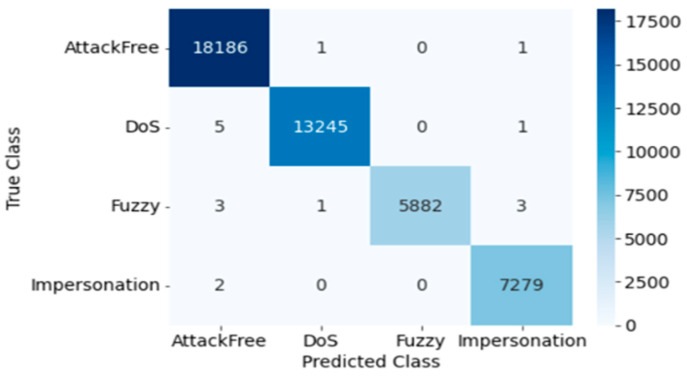
Confusion Matrix of DT (Dataset 2).

**Figure 11 sensors-23-03610-f011:**
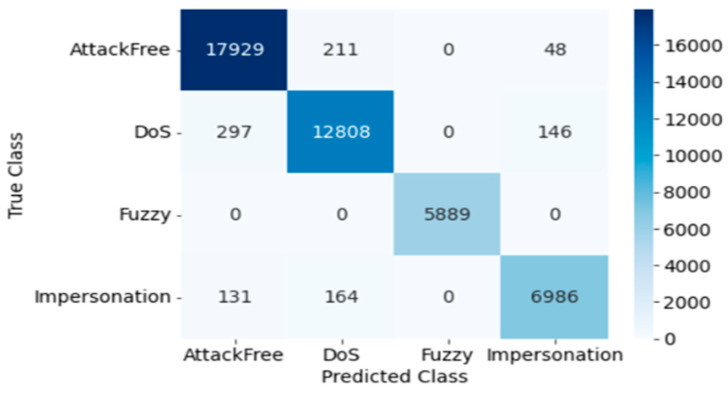
Confusion Matrix of KNN (Dataset 2).

**Figure 12 sensors-23-03610-f012:**
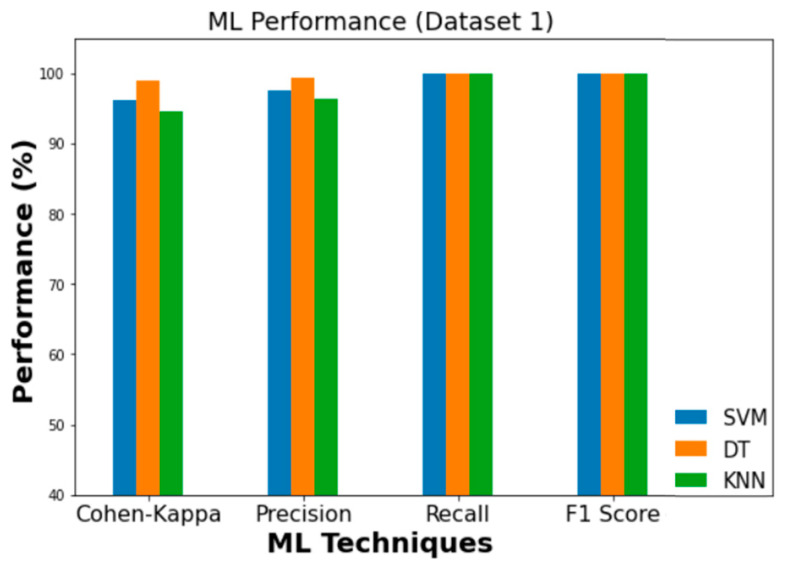
Performance comparison of ML (Dataset 1).

**Figure 13 sensors-23-03610-f013:**
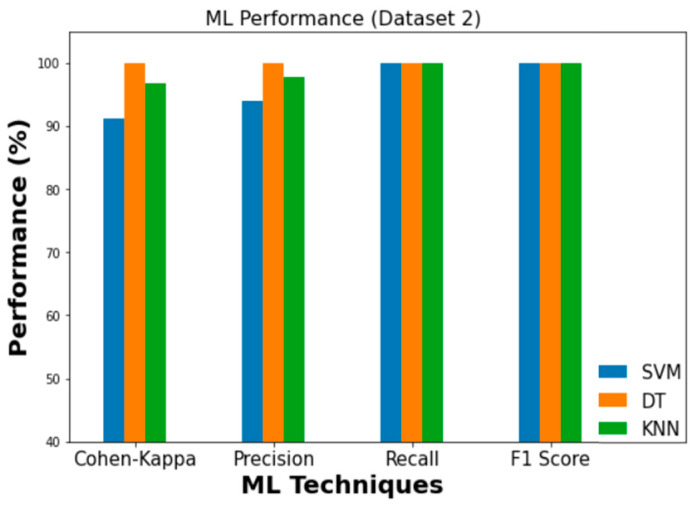
Performance comparison of ML (Dataset 2).

**Figure 14 sensors-23-03610-f014:**
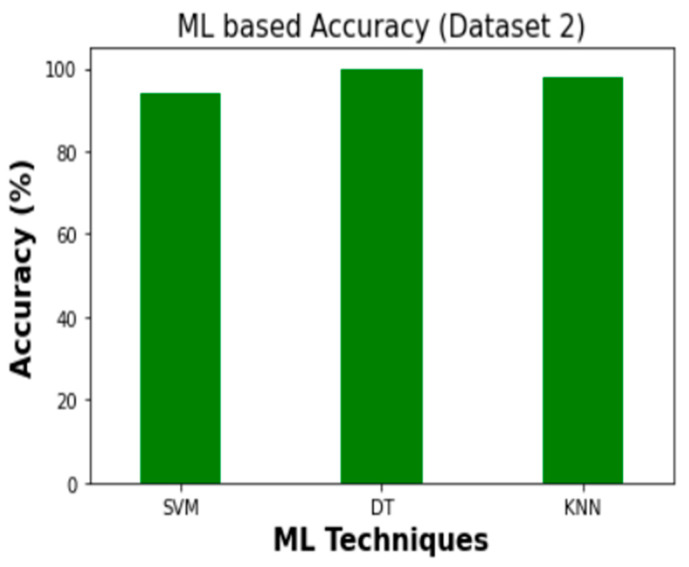
Accuracy comparison (Dataset 1).

**Figure 15 sensors-23-03610-f015:**
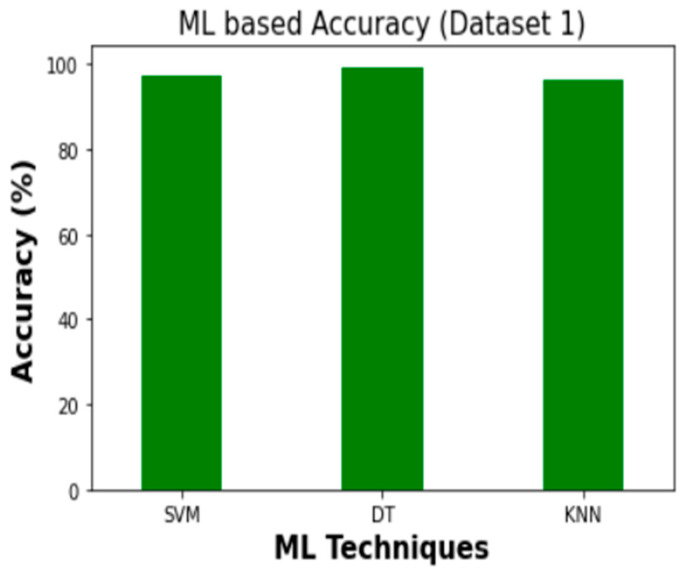
Accuracy comparison (Dataset 2).

**Figure 16 sensors-23-03610-f016:**
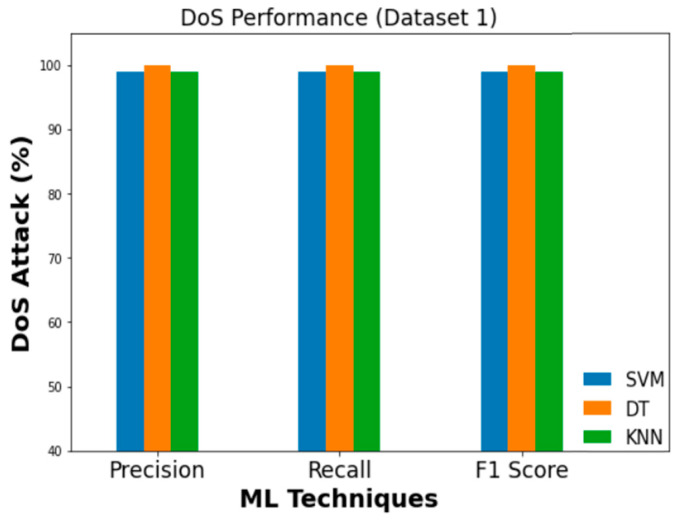
Performance evaluation metrics (DoS).

**Figure 17 sensors-23-03610-f017:**
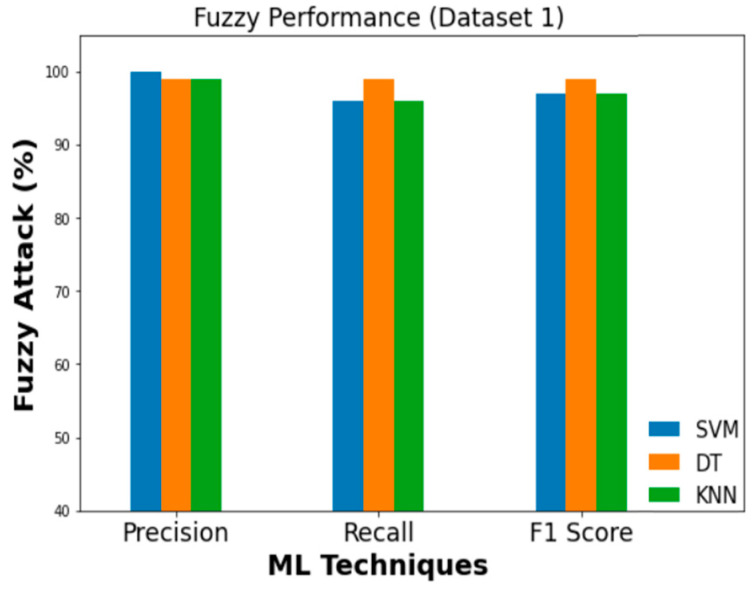
Performance evaluation metrics (fuzzy).

**Figure 18 sensors-23-03610-f018:**
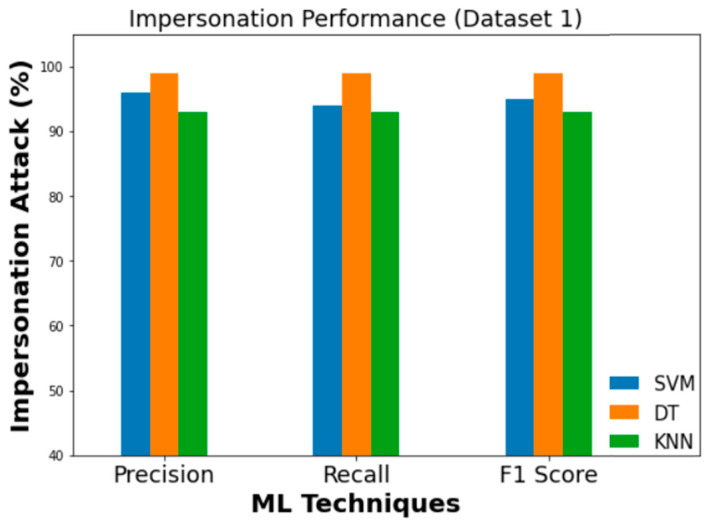
Performance evaluation metrics (impersonation).

**Figure 19 sensors-23-03610-f019:**
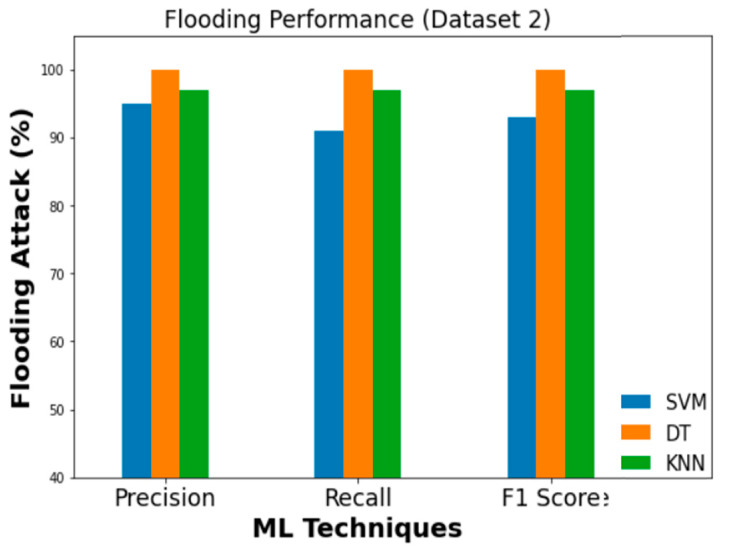
Performance evaluation metrics (flooding).

**Figure 20 sensors-23-03610-f020:**
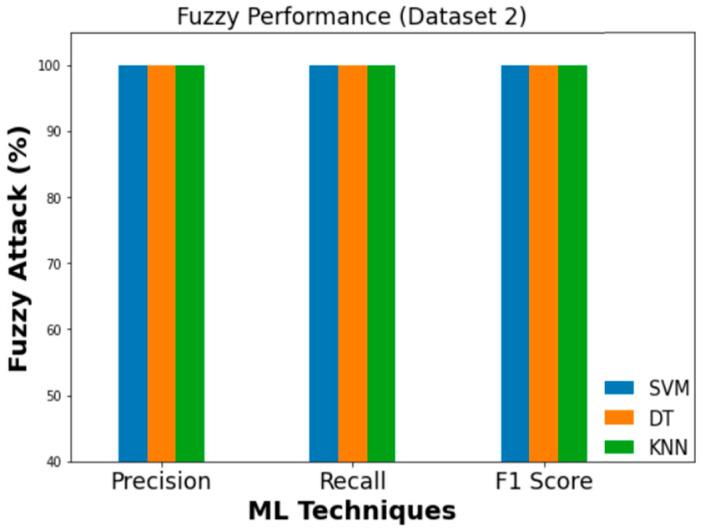
Performance evaluation metrics (fuzzy).

**Figure 21 sensors-23-03610-f021:**
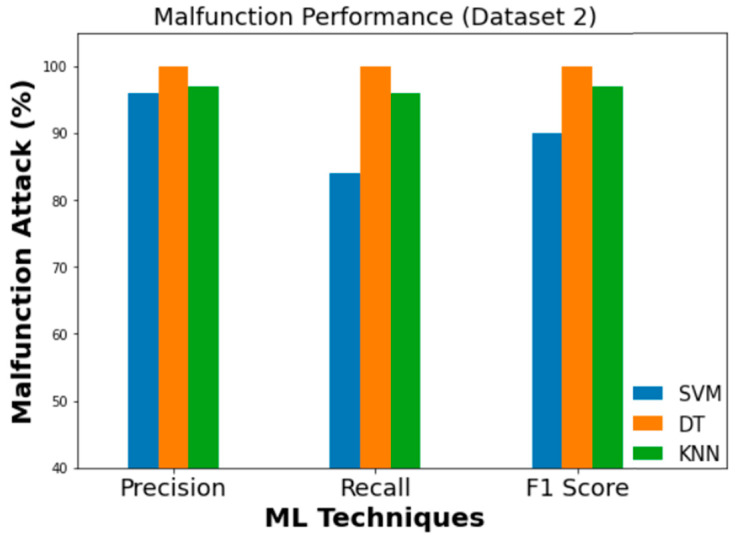
Performance evaluation metrics (malfunction).

**Table 1 sensors-23-03610-t001:** Summary of the state-of-the-art of intrusion detection techniques in vehicle.

Reference	Contribution	Method	Attacks	Impact Device	Strength	Limitation/Research Gap
Moulahi et al. [21], 2021	Four ML approaches for attack detection	RF, SVM, MLP, DT	DoS, impersonation, fuzzy	CAN	Real-time application in KIA Soul car	Large amount of dataset should be used
Liu et al. [14], 2021	Protect CAVs against perception error attacks	Periodic event-triggered interval of the CAN message	Flooding, fuzzy, malfunction, and replay	CAN	Real-time application	Tree-based ML model’s accuracy should be increased
Lee et al. [11], 2018	Decide if a behavior is an attack or a normal behavior	Remote frame and data frame to create an IDS	DoS, fuzzy, and impersonation	CAN	Detects the most dangerous attacks for vehicles	It does not analyze the accuracy of attack detection to determine whether or not the proposed approach achieved the best detection performance
Tariq et al. [45], 2018	Network traffic signatures and NN implementation	RNNs and heuristics	DoS, fuzzy, and replay	CAN	High accuracy (99%)	This system should be applied for unknown attacks
Miller et al. [46], 2015	Ability to hack the car anywhere in the US	-	-	Steering and brakes	Remote car hacking	This approach is required to be validated by applying the experiment to the new and updated vehicles
Groza and Murvay [15], 2019	Use bloom filter which is a memory-efficient mechanism	Data frame	Replay and modification	CAN	Real-time application with 100% recall performance	They included the overload on ECU, which could affect their time response
Jichici et al. [47], 2018	NN implementation to classify normal and abnormal activity	NN	Replay and injection	CAN	Replaying attack detection is hard due to the high degree of similarity between genuine frames and injected frames. It performed well in this case	Large memory requirements and computational time
Kang et al. [7], 2016	DNN implementation to classify normal and abnormal activity	DNN	Attack and non-attack	CAN	Simple and time efficient (2–5 mS for classification)	It is required to be applied in real-time application to validate the performance
Wu et al. [13], 2018	Use a fixed number of messages as sliding windows	Information entropy	DoS and injection	CAN	Reduce automotive costs and computing performance	The impact of the vehicle operation state on information entropy is not considered

**Table 2 sensors-23-03610-t002:** Key Features extracted from CAN vehicle datasets used in ML classifications.

Feature Number	Feature	Significance and Explanation
1	f1	Time stamp
2	f2	Last time stamp of remote frame
3	f3	Frame ID
4	f4	Previous frame ID
5	f5	Id of previous of previous of frame ID
6	f6	ID of previous of previous of previous of frame ID
7	f7	Data size in the frame
8	f8	First data byte
9	f9	Second data byte
10	f10	Third data byte
11	f11	Forth data byte
12	f12	Fifth data byte
13	f13	Sixth data byte
14	f14	Seventh data byte
15	f15	Eighth data byte

**Table 3 sensors-23-03610-t003:** Detection performance of DoS, fuzzy, impersonation, and attack-free for SVM (Dataset 1).

Attack	Precision	Recall	F1 Score	Samples (Testing)
No Attack	0.97	0.99	0.98	59,418
DoS	0.99	0.99	0.99	16,212
Fuzzy	1.00	0.96	0.97	14,699
Impersonation	0.96	0.94	0.95	25,007

**Table 4 sensors-23-03610-t004:** Detection performance of DoS, fuzzy, impersonation, and attack-free for DT (Dataset 1).

Attack	Precision	Recall	F1 Score	Samples (Testing)
No Attack	1.0	1.0	1.0	59,418
DoS	1.0	1.0	1.0	16,212
Fuzzy	0.99	0.99	0.99	14,699
Impersonation	0.99	0.99	0.99	25,007

**Table 5 sensors-23-03610-t005:** Detection performance of DoS, fuzzy, impersonation, and attack-free for KNN (Dataset 1).

Attack	Precision	Recall	F1 Score	Samples (Testing)
No Attack	0.97	0.97	0.97	59,418
DoS	0.99	0.99	0.99	16,212
Fuzzy	0.99	0.96	0.97	14,699
Impersonation	0.93	0.93	0.93	25,007

**Table 6 sensors-23-03610-t006:** Detection performance of flooding, fuzzy, malfunction, and attack-free for SVM (Dataset 2).

Attack	Precision	Recall	F1 Score	Samples (Testing)
No Attack	0.91	0.98	0.94	18,188
Flooding	0.95	0.91	0.93	13,251
Fuzzy	1.00	1.00	1.00	5889
Malfunction	0.96	0.84	0.90	7281

**Table 7 sensors-23-03610-t007:** Detection performance of flooding, fuzzy, malfunction, and attack-free for DT (Dataset 2).

Attack	Precision	Recall	F1 Score	Samples (Testing)
No Attack	1.0	1.0	1.0	18,188
Flooding	1.0	1.0	1.0	13,251
Fuzzy	1.0	1.0	1.0	5889
Malfunction	1.0	1.0	1.0	7281

**Table 8 sensors-23-03610-t008:** Detection performance of flooding, fuzzy, malfunction, and attack-free for KNN (Dataset 2).

Attack	Precision	Recall	F1 Score	Samples (Testing)
No Attack	0.98	0.99	0.98	18,188
Flooding	0.97	0.97	0,97	13,251
Fuzzy	1.0	1.0	1.0	5889
Malfunction	0.97	0.96	0.97	7281

**Table 9 sensors-23-03610-t009:** A performance comparison of different ML techniques and datasets.

ML Techniques	Precision	Recall	Accuracy	F1 Score	Cohen’s Kappa Score	Training Time (s)	Testing Time (s)	Dataset
SVM	0.975	1.0	0.975	1.0	0.961	1624	187	Dataset 1
DT	0.994	1.0	0.994	1.0	0.990	3.07	0.01	
KNN	0.964	1.0	0.964	1.0	0.945	0.044	300	
SVM	0.939	1.0	0.939	1.0	0.912	964	93	Dataset 2
DT	0.999	1.0	0.999	1.0	0.999	1.1	0.009	
KNN	0.977	1.0	0.977	1.0	0.968	0.02	144	

**Table 10 sensors-23-03610-t010:** Variation of true positive and false negative rates.

ML Techniques	True Positive Rate	False Negative Rate	Dataset
SVM	0.975	0.025	Dataset 1
DT	0.994	0.006	
KNN	0.964	0.036	
SVM	0.939	0.061	Dataset 2
DT	0.999	0.0003	
KNN	0.977	0.022	

**Table 11 sensors-23-03610-t011:** A comparison of the proposed work with existing techniques for intrusion detection.

	ML Techniques	Precision (%)	Recall (%)	Accuracy (%)	F1 Score	Cohen’s Kappa Score	Training Time (s)	Testing Time (s)	Total Data
Proposed Work	SVM	97.5	100	97.5	1.0	0.961	1624	187	461,341
	DT	99.4	100	99.4	1.0	0.990	3.07	0.012	
	KNN	96.4	100	96.4	1.0	0.945	0.044	300	
	SVM	93.9	100	93.9	1.0	0.912	964	93	313,930
	DT	99.9	100	99.9	1.0	0.999	1.1	0.009	
	KNN	97.7	100	97.7	1.0	0.968	0.02	144	
Moulahi et al. [21]	SVM	97.28	96.55	97.28	-	-	460.383	14.919	47,519
	DT	98.19	98.16	98.19	-	-	460.719	14.935	
Refat et al. [32]	SVM	98.61	96.09	97.92	97.26	-	-	-	56,256
	KNN	98.95	96.23	97.99	97.37	-	-	-	

## Data Availability

The GitHub link for our vehicular intrusion detection system (IDS) code/algorithm and associated documents is given in: https://github.com/BiftaSama/IDS_Sensor (accessed on 29 March 2023). Dataset 1: “HCRL—CAN-intrusion-dataset (OTIDS)” https://ocslab.hksecurity.net/Dataset/CAN-intrusion-dataset (accessed on 16 January 2023). Dataset 2: “HCRL-In-Vehicle Network Intrusion Detection Challenge” https://ocslab.hksecurity.net/Datasets/datachallenge2019/car (accessed on 16 January 2023).

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
