# Peer review of "Intrusion Detection in Vehicle Controller Area Network (CAN) Bus Using Machine Learning: A Comparative Performance Study"

_sensors, 2023, doi:10.3390/s23073610_

Round 1

Reviewer 1 Report

The abstract the author give us is totally not a abstract, please rewrite it to let me clearly and easily to access your work, novel idea, and your finding proposed in this paper, instead of just presenting the result and conclusions.

If this paper is a survey, it is not enough, there are large number of machine learning methods, not just SVM KNN and DT. 

If there is a novel method proposed, I see nothing new. 

Reviewer 2 Report

The paper is interesting, well planned and documented. The literature survey, contributions, methodology, limitations, and conclusions are well written. However, the work seems to be routine and lacks novelty, unless the following issues are addressed. 

1. Since the IDS is supposed to be working inside the vehicle, what is the point in using a desktop for conducting the research. 

2. It is highly recommended that you probe/tap the CAN bus and using an embedded processor simulate the attacks and test the results and performance. 

Round 2

Reviewer 1 Report

 make figures clear and don't let tables appear over two pages.

Author Response

Thank you for pointing out this. In the revised manuscript, we have provided more clear figures and addressed the issues regarding the tables.

Reviewer 2 Report

I agree with your response. 

You may include the following MDPI journal paper:

Swarm Optimization and Machine Learning Applied to PE Malware Detection towards Cyber Threat Intelligence

Author Response

Thank you very much for the valuable suggestion. In the revised manuscript, we have included this journal paper as a reference [25].

Reference:

[25] Perakovi, D.; Delia Jurcut, A.; Markovi, G.; Jhansi Kattamuri, S.; Kiran Varma Penmatsa, R.; Chakravarty, S.; Sai Pavan Madabathula, V. Swarm Optimization and Machine Learning Applied to PE Malware Detection towards Cyber Threat Intelligence. Electron. 2023, Vol. 12, Page 342 2023, 12, 342, doi:10.3390/ELECTRONICS12020342.